# Smart Nematic Liquid Crystal Polymers for Micromachining Advances

**DOI:** 10.3390/mi14010124

**Published:** 2023-01-01

**Authors:** Sébastien Dominici, Keynaz Kamranikia, Karine Mougin, Arnaud Spangenberg

**Affiliations:** 1Institut de Science des Matériaux de Mulhouse (IS2M), CNRS–UMR 7361, Université de Haute-Alsace, 15 rue Jean Starcky, 68057 Mulhouse, France; 2Université de Strasbourg, 67000 Strasbourg, France

**Keywords:** nematic liquid crystal, two photons polymerization, micro-machines

## Abstract

The miniaturization of tools is an important step in human evolution to create faster devices as well as precise micromachines. Studies around this topic have allowed the creation of small-scale objects capable of a wide range of deformation to achieve complex tasks. Molecular arrangements have been investigated through liquid crystal polymer (LCP) to program such a movement. Smart polymers and hereby liquid crystal matrices are materials of interest for their easy structuration properties and their response to external stimuli. However, up until very recently, their employment at the microscale was mainly limited to 2D structuration. Among the numerous issues, one concerns the ability to 3D structure the material while controlling the molecular orientation during the polymerization process. This review aims to report recent efforts focused on the microstructuration of LCP, in particular those dealing with 3D microfabrication via two-photon polymerization (TPP). Indeed, the latter has revolutionized the production of 3D complex micro-objects and is nowadays recognized as the gold standard for 3D micro-printing. After a short introduction highlighting the interest in micromachines, some basic principles of liquid crystals are recalled from the molecular aspect to their implementation. Finally, the possibilities offered by TPP as well as the way to monitor the motion into the fabricated microrobots are highlighted.

## 1. Introduction

Autonomous micro-actuators [1] have attracted the fascination of many researchers over the last few years for possible applications in the future. Micro-devices that can swim in various media [2,3,4], jump [5], crawl [6,7], or roll [8,9] on hard surfaces upon an adapted external stimulus spring from the imaginations of scientists for a world where injuries and diseases could be healed [10,11], oceans depolluted, or electric circuits could be even smaller. The inspiration for the larger part of the publications comes from nature itself reproducing the crawling of worms, opening of flowers [12,13], swimming of fishes, etc. Promising works have emerged in various fields such as drug delivery [14], in vitro fecundation [15], and robotics [16] where the materials were chosen to respond to a magnetic field [17], electric field [18], or even a light source [19]. Hard materials were commonly used but are not suitable for precise motion at the micron scale. Some materials such as polymers, along with hydrogels [20] or elastomers are soft and robust materials that can be tuned to be responsive and change their properties over time. Polymers’ biocompatibility offers a broad range of sensitive applications that can be coupled to biodegradability for their use in nature. They are generally low-cost materials but also easy to process and some are responsive to external stimuli. Finding the best material for micromachining design is a complex task where many directions are taken. Some favor pH activation [21] or magnetic activation, while others still favor light activation. Light activation is cheap and easy to use in a wide range of wavelengths making it a primary choice of the stimulus source. Liquid crystal polymers (LCPs) are sensitive to light, which makes them favored for their smart polymeric capacities. Nowadays, the developments around microstructures made of liquid crystal (LC) is of great interest in the community as we try to increase their programmability [22,23,24,25].

This review shows the great advances made in the LC micro-actuator field from molecular programming to activation with the different methods developed by researchers in recent years. We will focus on nematic LCs behaviors through a short introduction. Then, existing alignment techniques will be highlighted through several works in this field. Next, different actuation protocols will be described for nematic LC matrices. Furthermore, a great part will be devoted to the microfabrication of LC objects emphasizing the Two-Photon Polymerization (TPP) method, which will be demonstrated through the great majority of the examples shown to demonstrate the fidelity and originality of this technique. A further section will describe the latest works on the actuation of LC micro-structures with different stimuli or methods. Finally, a short section will cover the color actuation of LC structures.

## 2. Alignment Method of Liquid Crystal Mesogens

LC responsiveness to stimuli was discovered in 1888 by F. Reinitzer [26] and characterized also by O. Lehmann [27]. An important factor to control the responsiveness of an LC material to several stimuli is its molecular alignment quality, which has an important impact on the anisotropic properties of the LC matrix. A few techniques to control the directions at the molecular level already exist [28] and will be described for nematic LC, particularly for 3D printing. 

Nematic LCs have a particular behavior created by the anisotropic shape of the molecules due to their longitudinal form as Figure 1 illustrates. In their solid form, they have a crystalline conformation with a position order of the molecules, and, when heated to their liquid form, a nematic state, also called a mesophase, is obtained. In this state, the molecules have an orientation along a direction n but several microdomains can appear creating many directions. The range of temperatures across which the nematic phase persists can be more or less extended through the use of different resins and heating further will lead to a transition temperature called the nematic-isotropic temperature. This transition state corresponds to the moment that the LC molecules lose their alignment to their isotropic configuration. The point at which the resin goes from a blurry state to a very transparent state is visually observable and is called the clarification point. On cooling the sample, a nematic state can be obtained once again and crystallization will occur when the crystallization temperature is reached. At the macroscale, a global orientation can be induced by applying a constraint over the LC resin. The creation of LC cells is commonly used where the top and bottom substrate surfaces are previously treated with a PI or PVA polymer that would be rubbed by a cloth creating furrows [29] as shown in Figure 2A. These two polymers were chosen for their excellent film forming, good thermal stability, and chemical resistance. They are also easy to use, with spin-coating being able to control their film thickness and mechanical force able to modify their surface. They also have good adhesive properties. Once the surface is dry, LC resin is introduced in the cell at an isotropic temperature by capillarity and then cooled down to RT. The molecules will follow the furrows creating a global mesogenic orientation that can be parallel or perpendicular to the surfaces. Extrusion [30] Additive Manufacturing (AM), Fused Filament Fabrication (FFF), and Direct Ink Writing (DIW) are also techniques using mechanical forces that are well-known for allowing the orientation of LC molecules. Imposing a magnetic field [31,32] will also influence an extended zone of an LC resin and the field can be designed in 3D depending on the disposition of the magnets. As shown in Figure 2B, LC molecules will have a tendency to follow the magnetic field direction because of their magnetic anisotropy. A few studies report planar and non-planar orientation upon activation with an external stimulus. Some LC resins are sensitive to electric fields [33] (Figure 2C) and the molecules can follow the direction of the field. To use such a method the LC cell needs to be composed of two transparent electrode layers such as an indium tin oxide (ITO) replacing the glass substrate. The intensity of the current applied has a great influence on how the mesogens will rotate allowing control of the degree of orientation. The addition of a dopant such as an azobenzene [34,35], as shown in Figure 2D, offers another way to control LC alignment with polarized UV light. The azobenzene molecules will follow the light polarization direction and the LC molecules will be led in the same direction making them reconfigurable (Figure 2D). The azobenzene used can be one of several forms that have been optimized for LC alignment protocols [36,37,38]. Finally, the last example shown in Figure 2E shows a pre-patterning of two glass substrates [39], which will form the boundaries of the LC cell. The pattern can be of different shapes at the discretion of the operator to change the alignment as needed. The LC resin is then introduced by capillarity in its isotropic phase for the molecules to follow the pattern and a structure can be written with the desired orientation. It’s a more precise technique than those presented above as the patterns are composed of squares 10 microns in length. Applying a mechanical force such as stretching can also result in a single monodomain of LC [40]. However, the latter approach is less used these days. 

For the printing of micromachines, a few techniques exist such as two-photon polymerization (TPP), micro extrusion, and micro molding to keep an LC alignment. In this review, we mostly focus on works dealing with TPP as it is the more precise technique in use. Today, most of the LC orientation techniques have been used for TPP even if they are still to be improved. To obtain a structure with the desired properties, the first step is to understand the components of the basic photoresist. 

The two main components of a basic photoresist are, in general, a difunctional acrylate LC, bringing strength to the network once polymerized and keeping the LC molecular alignment, and a monofunctional acrylate LC that reduces the viscosity of the resin. The ratio between those two is crucial to tune the properties of the final polymer and how it will react upon activation. Too much difunctional monomer content will lead to a lack of motion freedom of the structure and too much monofunctional monomer content will lead to an unshaped structure with very low mechanical properties. In general, there is a window where we can work to reduce or enhance each property. Once found, the mixture should be stable at room temperature for long enough to fabricate the microstructures, which can last from a few seconds to a few hours. The use of a heating stage can avoid this limitation and allow the LC mixture to be heated to its nematic state. Another essential element when TPP is involved is the use of a photo-initiator compatible with the laser wavelength. For acrylate monomers, free radical generation is often used to initiate the reaction, followed by a propagation step where the polymer chain is formed and starts to grow, and, finally, the termination, meaning the end of the polymerization reaction. 

## 3. Actuation Techniques for LC Molecular Disorientation

Actuation in LCN polymers is directly correlated to its molecular alignment properties concerning the direction n, crosslinking, orientation quality, and design of the structures. For the majority of printed LC objects, a contraction of the network will occur along the molecular orientation direction n as the molecules will lose their alignment. They are linked to each other through their C-C bonds, which belong to their acrylate groups. An elongation of the material will appear perpendicular to n proportional to the contraction intensity. Different triggers allow the bending of LC molecules, as is shown in Figure 3. The most common trigger is heat [41,42], where the temperature can force the molecules to lose their alignment. It is the passage from the nematic phase which presents a global alignment of the molecules to the isotropic phase where a molecular disorder appears, also called T_N-I_. The structure will be deformed in a predictable way and the amount of heating will control the degree of torque imposed on the structure. Cooling down the sample would give the polymer its initial molecular order and also its first structural shape. The second most used stimulus is light actuation [43] through the addition of a dopant that can produce either a photomechanical or photothermal effect. 

Azobenzenes [44,45] with bifunctional acrylate groups can change from trans to cis conformation upon a certain wavelength in the UV area (mostly around 365 nm) and go back from cis to trans conformation after removing the irradiation. The structure undergoes an intrinsic mechanical pressure at the molecular level due to these changes, which will contract it along the alignment axis. Depending on the chemicals used, the transition can be more or less rapid (seconds, hours, days…) and some parts of the polymer can take more time than others to take back their initial form. This will bring differential stress, which can be mastered to create other movements such as a spring or a wave. Also, usually, the cis configuration shows a different emission wavelength than the trans configuration, and a few colors can then be noticed before and after light irradiation. The azobenzene photomechanical effect is commonly coupled with a photothermal effect.

Some dopants, such as gold nanoparticles [46,47], carbon nanotubes, dies [48,49], or monofunctional azobenzenes, can produce a photothermal effect by absorbing light energy and transforming some of it into heat. This increase in temperature will locally allow greater freedom of the molecules in the surroundings and, if incorporated into LC mixtures or polymers, will allow the LC to reach the T_N-I_, creating a disorganization of the molecules. By incorporating such dopants in an LC resin and, after alignment of the mesogens, polymerizing it, a structure responsive to light can be obtained. Several factors will be at play concerning the intensity of deformation such as the density of photothermal entities and the degree of crosslinking influencing the heat diffusion. The illumination wavelength can be tuned with the incorporation of different particles. For gold nanoparticles, almost the whole spectrum can be used for their light absorption by varying their size and shape. For example, a cylindrical gold nanoparticle can absorb both UV light corresponding to its tubular axis absorption and at 532 nm with the perpendicular axis absorption. Making the tubes longer allows the particle to absorb in UV and at 800 nm. Nowadays, using this theory, researchers try to create structures with multi-responses to several wavelengths. Light is a low-cost actuation but is also problematic because of its low penetration depth.

Another stimulus studied for many years is humidity [50]. To this end, the LC mesogens must be hydrophilic and show a certain degree of polarity. Polar groups such as carboxylic acids can interact with the water molecules and create weak hydrogen bonds. The degree of polymerization will have a great influence on the swelling of the structure proportional to the hardness of the material and the diffusion of the molecules inside the polymer network. Creating differential polymerization in a structure would result in various swelling intensities, and all kinds of movement are possible. A humidity chamber can be used to control the humidity around the structure as well as the drying. The swelling can also imply changes in color as the LC material thickness will change and will impact its emission behavior. 

An electrical stimulus could be used to induce a movement if the LC structure is already molecularly oriented but also by integrating dopant nanoparticles in the resist that will absorb the energy and transform it into heat. This could allow the T_N-I_ to be reached and deformation of the structure to be observed. This approach exists [51] in other domains but can be developed in activable LC micromachines.

The magnetic stimulus can have two possible effects. The first is a magnetic attraction where magnetic particles were included in the resist and the final structure will be responsive to a magnetic field. There are some studies showing large displacements with magnetic micro- and nano-beads. The second is to dope the polymer or its support with iron or a magneto-responsive metal that is well-blocked in the polymer network. The energy absorbed by the structure from the magnetic field will be transformed into heat as described in magnetic hyperthermia research [52]. Also, a strong magnetic field can induce at high temperatures a small reconfiguration of the molecular alignment after polymerization, changing the deformation direction.

## 4. Deformation Induced by Molecular Orientation Programming

For LC films in an aligned molecular order following a director axis n, upon activation, a contraction will occur along the n axis as illustrated in Figure 4. This happens when the sample is heated as the T_NI_ is reached causing misalignment of the LC molecules and their contraction. They will get closer to each other, reducing the length of the material in its aligned axis. In Figure 4A, the molecular orientation is designed to be parallel to the film’s long axis. By applying the stimulus heat to the samples, the film will be reduced in size along the Z-axis corresponding to its former orientational axis and thereby expand in the X- and Y-axes. Conversely, Figure 4B corresponds to an orientation of the molecules in the short axis perpendicular to the film’s long axis. Upon heating the film, a contraction will occur in X-axis and there will be an elongation in the Y- and Z-axes. By using this principle, the programming of films at the macroscale can be engineered to obtain a certain movement in any planar direction. A differential molecular alignment in a planar film can induce 3D deformation as is the case in splay orientation when the molecules are progressively twisted along the n-axis. This can lead to a bending of the film to obtain a roll or a twisting to obtain a screw-like form. In addition, the thickness can be modeled to differ across the same film to obtain different contraction forces and thereby a 3D movement. At the microscale, the molecular alignment is more difficult to control and, in practice, is made by orienting an extended zone of material. For 2D objects, their alignment will depend on the alignment of the substrate they are fabricated on or an external stimulus such as a magnetic or electric field. For 3D objects, the same rule applies as they will be constructed layer by layer. If the orientation depends on the substrates, the order will change while the fabrication gets higher in resin content. The contact with the substrate will align almost perfectly with the resin nearby but further away moving upwards the resin gets more and more disoriented. In this way, a differential in orientation can be induced. Another way to create several orientations in a 3D object is to change the direction of the stimulus—generally a magnetic or electric field—during the fabrication. These differences in orientation can module the deformation of the final objects.

## 5. Existing LC Micro-Object Fabrication Techniques through a Polymerization Approach

Different polymerization techniques have been developed over the years for microscale object resolution. Figure 5 shows all the different ways for micro-structuration based on the LC resin used. The first method (Figure 5A) begins with the preparation of a cell made with two glass slides and spacers of the desired size. Then a drop of LC resin is deposited near one side of the cell and will flow inside by capillarity aspiration. When the cell is full, the polymerization process is begun by irradiating with UV light at the wavelength at which the photo-initiator is degraded. After removing the glass, an LC film is obtained but the resolution is low, at around 10 µm. The second method, depicted in Figure 5B consists of the infiltration of the LC resin into a mold and, when the UV light hits the cell, only the regions where the resin could reach are polymerized. This is a more precise technique than the first with 1 µm resolution but requires transparent molds corresponding to the final structure, which can be time-consuming. The third method (Figure 5C), also called micro-extrusion, is a technique where a reservoir is filled with non-oriented LC resin. Then, the resin is heated to just under the T_NI_ to decrease the viscosity and ensured to be in the nematic phase where molecules are already in a pre-oriented state. The resin goes through the micronozzle, which will force the molecules to orient themselves in a single direction. After deposition, the resin is polymerized by UV light. This allows the writing of a structure in 3D but has a low resolution of around 100 µm. Another technique is to deposit a drop of resin on a glass slide and focus the light through a microscope objective. This greatly improves the resolution to 1 µm, and the resolution can be further improved with two-photon polymerization, as we will see later. 

In this review, we will focus largely on TPP [53], a more precise technique than any other to date and which allows working in an oriented LC resin at the same time. Figure 5D explains in detail all the features of this technique in the LC cell. First, a pre-alignment layer has to be coated on two glass slides using either PVA (PolyVinylAlcohol) or PI (PolyImide). Second, an LC cell needs to be made with these two glass substrates separated with spacers such as silicon micro-beads. The next step consists of the infiltration of the LC resin under its T_NI_ by capillarity in the cell giving the LC molecules a defined orientation. Next, a femtosecond laser emitting 120 fs pulses with an 80 MHz repetition rate, generally in a red color (780 nm), is focused in the resin. The focal point is called the voxel and the polymerization occurs only at this point as it is where the photon density is higher and degrades the photo-initiator, starting the polymerization. This technique is giving great freedom in 3D as the laser can travel freely inside the resin making any structure possible in theory. The microscope objective can be used to tune the size of the voxel. When the magnification is increased, the voxel size is decreased allowing better resolution and vice versa.

## 6. Micromachines Now

Various studies show LC microstructures with many types of deformations due to their molecular orientation, their design, or both. Many examples can be cited but a selection will be presented in this review focusing on the TPP technique. Currently, this topic of research is based on future possible applications for micro-medicine and micro-robotics, as current technology is not advanced enough to obtain extremely precise motion with LC robots at the microscale. In Figure 6A, LC cubes have been oriented in different directions using an LC cell with preorientation substrates made of glass covered with rubbed polyimide [54]. The rubbing direction of the polyimide defines the global orientation of the molecules and the structures were written in different ways; some were parallel to the director n, some were written diagonal to n and others were written in different 3D positions, as described in Figure 6A (i). This technique gives a large choice of possible orientations and, thus, deformations. The SEM images in Figure 6A (ii) and optical images in Figure 6A (v) show similar cubes, demonstrating that the designs are the same and were not influenced by their molecular orientation. In Figure 6A (iii), optical pictures of the cubes are shown before their exposure to high-temperature actuation (Figure 6A (iv)) where the deformation is visible and different. Once this was demonstrated, those cubes of 100 µm width were assembled in 3D layers to obtain different millimetric parallelepipeds with their own deformation. This assembling technique presents a great opportunity to design a wide range of structures with all kinds of deformation but is also long and difficult to put in place. In Figure 6B, a micro-walker device has been designed [55]. The upper part, which corresponds to the body, is made of LC polymer doped by an azobenzene molecule, and the legs are made of IP-Dip. Both parts have been written by TPP starting with the legs, then developing and adding the upper photo-responsive body part. Then, upon irradiation, a contraction occurred in the axis of the molecular LC alignment. With this, the authors were able to control the deformation and thus the movement of the micro-walker. Several movement patterns have been studied, from rotating to walking in a straight direction with an average speed of 37 μm s^−1^. The third example (Figure 6C) shows structures also aligned with an LC cell where the LC photoresist is infiltrated [56]. TPP is used to make two structures for optical and polarized characterization, the first one is a woodpile, and the second is a spiral disk. Both are 3D structures that are observed under polarized light with the elevation in temperature. While heating the structure from 30 °C to 100 °C, few changes are noticed. The woodpile appears blue and the spiral disk is a mix of red, blue, green, and pink at the center. From 100 to 150 °C, the edges of the woodpile became redder, indicating that there is a contraction inside the material, and the spiral disk goes from pink to green at the center. Then the temperature is increased further to 200 °C and the structures show great changes. The woodpile becomes redder as does the spiral disk, except its center, which has become blue. Furthermore, structural changes are also visible in addition to color changes. Indeed, a contraction is observed along both structures’ molecular alignment direction. This contraction when the T_NI_ is reached leads to an expansion perpendicular to the alignment axis. This example shows great freedom of design where a similar deformation can be obtained from strictly different structures and thus can be adapted to the targeted application. Kirigami structures [57] have also been studied by various groups, and specifically using LC material, by Sitti and al. (Figure 6D). First, LC resin was infiltrated between two glass substrates. At this point, the resin in the nematic phase shows many domains of orientation, and to align them all, a stretch of 150% was performed resulting in an oriented LC layer. The LC was then frozen in this state by exposure to UV light, which crosslinked the resin into a hard LC polymer. A solid monodomain layer was obtained and substrate squares were made from this by cutting with a laser. Then TPP at 780 nm was used to write on the top of the squares with IPS780 photoresist and make kirigami structures. The unreacted resin was then removed with isopropanol revealing structures similar to a bridge but constituted with two anchors at each side sustaining a long plateau. This plateau itself is composed of numerous squares linked to each other by their tip end and that can be redesigned in different ways to obtain multiple deformation behavior subsequently. By heating the LC substrate, an expansion happens orthogonally to the director n and the anchors are pulled apart from each other. The first squares linked to the anchors drag the others through their linked tips creating a mosaic. This study shows the power of LC material not only as the structure itself but how it can be used as a substrate to empower usually unresponsive hard materials. This is an opportunity to use the characteristics of LC actuation and add other properties to the structures such as the hardness or even the capacity to swell. Kirigami techniques have already shown great potential and could be in the future applied directly to LC structures. Other studies were driven to obtain a defined LC molecular alignment at the microscale.

Wegener and coworkers [58] (Figure 7A), among other research groups [59], described a new technique to orientate the LC mesogens during fabrication in 2D and 3D to create responsive micro-objects with TPP. The LC mixture is infiltrated between an electrode at the bottom and a substrate at the top. Four electrode pads are placed in the cell to create a 2D alignment and two are placed on the outer part of the cell to induce a current in the Z-axis and, thus, a molecular alignment. If the current is applied between two electrode pads inside the cell, the electric anisotropy of the LC mesogens will force them to be oriented in the current direction. The same mechanism applies for the electrodes on the outer part of the cell, inducing an orientation in the Z direction. Proof of concept has been demonstrated by fabrication in 2D, then in the Z-axis of membranes that have been observed under polarized light. The orientation demonstrated that a structure was written with one plot basis where eight bilayer beams were attached. One side of each bilayer was oriented in a different direction and the beams were written with a torsion. Upon activation, the beams twist in the other direction creating a helix movement. To go further with this technique, another structure was made with two plots acting as supports and two beams connecting one to the other. Each section of the beams is made by a bilayer where the LC layer has an orientation and the second layer does not. By heating the structure, a movement is observed until cosine-shaped beams are obtained. The transition appears quickly at 10 °C, leading to a shape reconfiguration. This is due to three different reasons. The first is the alignment of the LC layer, which promotes a contraction in a certain direction where this movement can happen. The second relies on the LC material becoming softer when heated and the other layer less affected by heat keeping its hardness and forcing the second layer to bend the other way. The third resides in conformational stability, which explains the amplitude of movement and shows that, by exploiting this property, one can create intense deformation. To obtain a greater movement or obtain another actuation method, researchers also exploit dopants such as azobenzene molecules or gold nanoparticles. In Figure 7B, a study by Xiaolin Xie et al. [60] deals with the use of gold nanoparticle dopants inside an LC mixture to create a photothermal effect upon illumination with near-infrared light and the deformation of their structures. The orientation of the LC molecules was achieved through an LC cell with a PI coating and the infiltration of the resin. The resin itself was doped with gold nanoparticles having the ability to absorb at NIR wavelengths and then microstructures were printed inside this with TPP and an ×63 microscope objective. Once the crosslinking finished, the resin was removed and some gold nanoparticles were stuck into the network. Then, illumination with NIR light where the gold nanoparticles were absorbed is performed and results in a photothermal effect allowing the LC to reach the T_NI_. A contraction happens along the n axis when the molecules are disoriented and, when the exposure to NIR light stops, the structure takes back its first shape. This is a doped and reversible way to create motion in an LC polymer. The implementation of gold nanoparticles is nonetheless limited because they disturb the laser while writing and usually cause the burning of the structure. For light actuation, azobenzene molecules or photothermal-responsive colorants are also used, and generally for NIR light because the light penetration into polymer or tissues is better at these wavelengths. Another recent study on micro-actuators activated with light comes from Blasco et al. [61] (Figure 7C) where micro-clamps were developed. Here, a two-step method is presented that begins with the fabrication of a micro-actuator in a pre-aligned LC cell that is immersed in a dye solution. The dye will integrate the structure and, after development, the right illumination wavelength will activate the structure and, in this case, open the clamp. The interesting part resides in the multiwavelength activation allowing several structures next to each other to be had and, upon illumination, selectively open the desired clamps. The authors also show the repeatability of the results and the cyclicity of the reaction when the light is on or off. 

Many other approaches have been developed in recent years on LC microstructures in optics [62], robotics, and many other fields. Research allowing the doping of LC structures with magnetic microparticles [63,64] or nanoparticles has also been pursued by many scientists for their multiple properties and possible application in medicine. As detailed in the next section, LC structures can also have great properties over color changes when observed with polarized light, facilitating the observation of the reconfiguration of the polymer network at the molecular level (Figure 8C).

## 7. Color Change Study on Microstructures

LC structures have the characteristic of shrinking along their molecular orientational axis under different stimuli and these changes are observable both structurally by optical microscopy and by their change of color under polarized light. A change in the thickness of LC objects will change the refractive index of the structure and its emission spectrum. Figure 8A explains how temperature induces a reduction in the thickness of a parallelepiped and subsequently how it impacts the color of the microstructure. At T_0_, which corresponds to room temperature, the parallelepiped is blue, but increasing the temperature to T_1_, will shrink the structure and turn it green. Increasing the temperature to T_2_ > T_1_ will again change the color from green to red. Since the maximum shrinkage is at T_2_, if the temperature is increased again no change of color will be observed. This characteristic has been demonstrated in numerous studies with LC films that are either stretched mechanically or shrunk by an external stimulus. Both of these techniques change the thickness of the films and induce a color change. This is demonstrated by Schenning et al. [65], as shown in Figure 8B, with thermotropic LC films where the alignment was achieved with a PVA-coated LC cell and fixed by UV light polymerization. Observation under polarized light has been performed at room temperature and the film appeared blue. Then, upon heating the film to 171 °C, a shrinkage along the director n and a change of thickness has been detected as the color changed to red. Furthermore, the article describes how to design a 3D beetle in from a 2D film that will be blue, where, upon activation, the beetle will flatten in a 2D film again and become red. This phenomenon is also well described in other publications [66,67]. Some groups, such as Sitti et al. [68], applied this strategy to micrometric structures, as demonstrated in Figure 8C. At this scale, it has been less described and has a great potential for sensor application or micro-robot detection. Heat was used as the source of activation as other techniques may be difficult to use at this scale. A pyramidal structure 10 microns high has been made and a different color is noticeable at each step. Likewise, squares of different thicknesses were fabricated next to each other to observe a gradual color change. The temperature was elevated from 23 °C to 130 °C with pictures taken at 70 °C and 100 °C in between. A clear color change was noticed and the idea of applying this phenomenon to microrobots and detecting their deformation through their color emission is mentioned. Other designs are studied by other groups with different actuation methods such as the work of Elston et al. [69] shown in Figure 8D. The molecular orientation was achieved through a coated LC cell with substrates made of ITO and the structures were fabricated by TPP. After development, a set of pillars were obtained that could be observed by optical or polarized light. They noticed that the refractive index could be changed by applying different voltages. By taking advantage of this observation, a reconfigurable smiley or shield was demonstrated (Figure 8D). We can see great changes in the shape and color while the current was increased due to the change in the refractive index. When the pillars are disappearing, this means that the voltage used to read the pattern is the same as the one used to write it. This work shows possible applications for LOGOs or passcode reading as well as art design. Other works have been submitted using an electric current to design pixels [70]. Many other works are based on other actuation systems that can influence color changes. This is the case in swelling structures [71] where the thickness is changed by swelling with water absorption and a clear color change can be observed. This non-destructive technique also requires very little energy compared to thermal or electrical actuators and is reversible. It can also be an anisotropic deformation regarding the anisotropic conformation of the molecules themselves.

## 8. Conclusions

Micromachining is a topic of primary interest for the miniaturization of tools in a variety of fields. As reported in this review, numerous recent studies involving 3D microfabrication via two-photon polymerizations have emphasized the ability to sculpt the matter with an unprecedented degree of freedom. Liquid-crystal polymers have been envisioned as the material of choice to achieve the production of complex 3D microstructures exhibiting controlled deformations on demand. Interestingly, while advanced 3D printers have been involved in the fabrication of such structures, most of the works can be summarized as an adaptation of results obtained from the macroscale to the microscale. However, several aspects will have to be addressed in the near future in order to achieve the production of micro-robots presenting various types of deformations in response to one or several stimuli. Among the various parameters, the orientation method is undeniably the key to unlocking advances in this area. Thus, the most popular way to orient LC molecules is the use of an LC cell coated with a pre-alignment polymer. While this methodology has been successfully used to make reconfigurable thin films, its use in combination with TPP drastically reduces the degree of complexity that one can expect when coupling TPP and LC polymers. More interesting is the use of an electrical field to orient the LC molecules into a hierarchical 3D structure. While impressive deformations have been demonstrated, orientations of LC molecules at that scale require strong electrical fields and bulky equipment. In addition, multi-orientations are limited due to the time required to tune the direction of the applied electrical field. To conclude in this aspect, although different orientation methods have been proposed to order LCs into microstructures, the proper orientation method should be selected considering the targeted application and trade-off between advantages and limitations of each method (resolution, multi-orientations to allow several types of motion in a single robot, orientation time). It should be mentioned in this context that other directions have to be explored to reach optimized micro-robots including (i) material selection (e.g., propensity to self-organize or in response to given stimuli, reactivity, mechanical properties), (ii) appropriate design (the design should favor new modes and amplitudes of deformation), (iii) fabrication and orientation methods (resolution, production time, multi-material capability), (iv) actuation methods (multi-stimuli, speed, resolution), (v) Real-time visualization of the deformation. Concerning the latter point, the structural color arising from the organization of the LC polymer chains is an appealing way to visualize the reconfiguration or motion of 3D micro-robots when subjected to a given stimulus. Finally, in order to translate microrobots from the laboratory to real applications, performance (e.g., exerted force, speed, control of the motion, kind of deformation) should be considered and systematically investigated.

## Figures and Tables

**Figure 1 micromachines-14-00124-f001:**
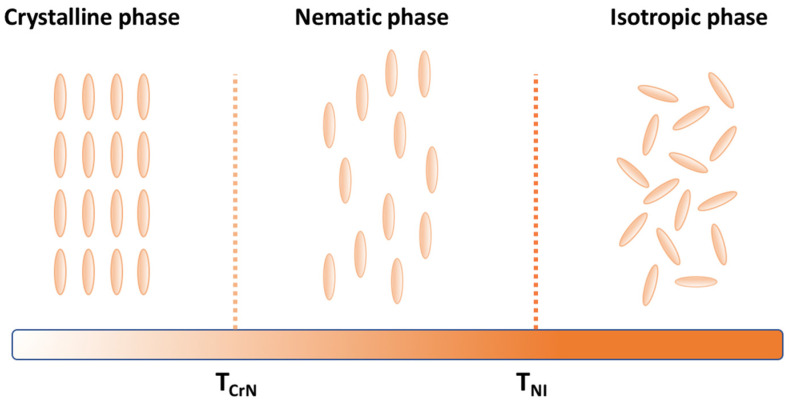
Phase transition behavior for a nematic LC with an increase in temperature.

**Figure 2 micromachines-14-00124-f002:**
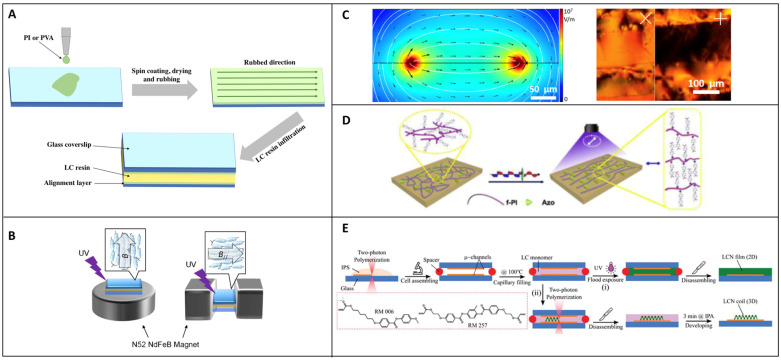
(**A**) Scheme of an alignment with substrate-coated LC cell. (**B**) Magneto-responsive mesogenic orientation [31]. Copyright © 2016 American Chemical Society. (**C**) Electric field-driven LC molecular orientation [33]. Copyright © 2021 Carlotti M et al. (**D**) Doped LC resin with UV-responsive azobenzene [34]. Copyright © 2018 Elsevier Ltd. (**E**) LC molecular orientation via repatterned glass substrate with two-photon polymerization [39]. Reproduced under the terms of the Creative Commons CC BY license. Copyright 2020, Wiley-VCH.

**Figure 3 micromachines-14-00124-f003:**
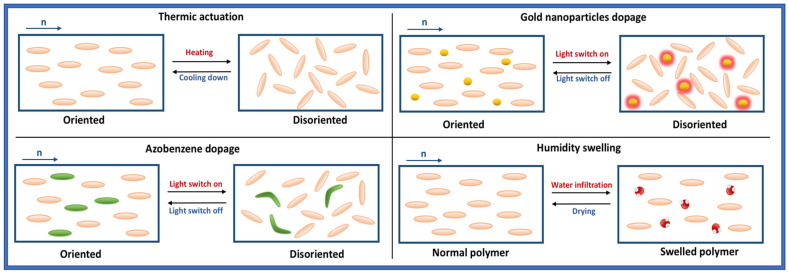
Existing actuation methods with a thermal, light, or humidity stimulus.

**Figure 4 micromachines-14-00124-f004:**
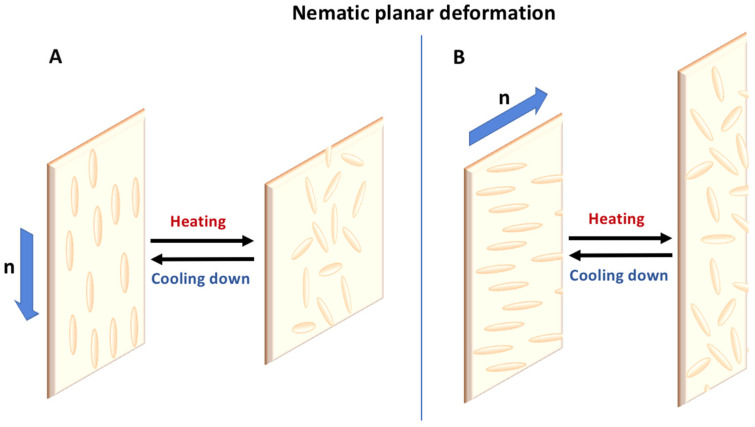
(**A**,**B**) Planar deformation with orientation programming of LC molecules.

**Figure 5 micromachines-14-00124-f005:**
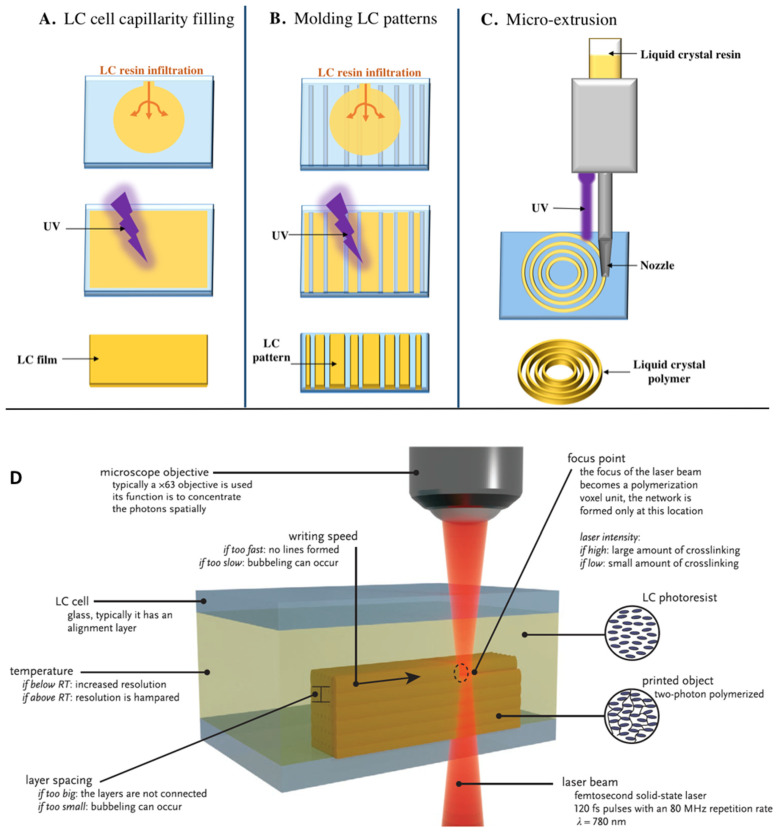
(**A**–**C**) Different techniques for the impression of objects at the micro-scale involving a photopolymerization reaction. (**D**) TPP applied to an LC pre-oriented cell [53]. Reproduced under the terms of the Creative Commons CC BY license. Copyright 2021, Wiley-VCH.

**Figure 6 micromachines-14-00124-f006:**
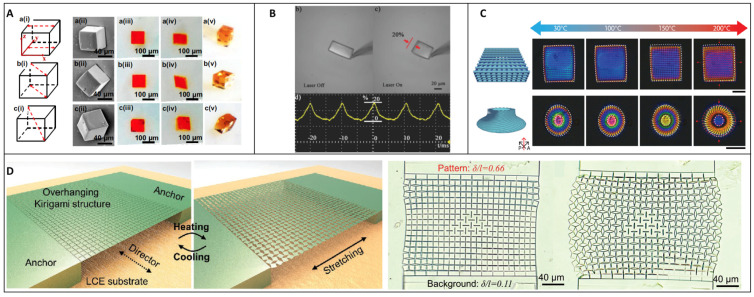
(**A**) Micrometric blocks with various orientations for millimetric-responsive structure construction [54]. Reproduced under the terms of the Creative Commons CC BY 4.0 license. Copyright 2021, Nature communications. (**B**) Photo-responsive micro-walkers [55]. Reproduced under the terms of the Creative Commons CC BY-NC 4.0 license. Copyright 2015, Wiley-VCH. (**C**) Micro-structuration of a woodpile and a spiral disc with observed expansion under thermal actuation [56]. Reproduced under the terms of the Creative Commons CC BY 4.0 license. Copyright 2021, Wiley-VCH. (**D**) Kirigami structure constructed on an LC film thermally actuated [57]. Reproduced under the terms of the Creative Commons CC BY 4.0 license. Copyright 2021, Wiley-VCH.

**Figure 7 micromachines-14-00124-f007:**
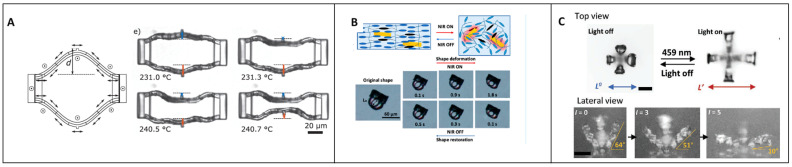
(**A**) Thermal actuation of an electrically aligned micro-object [58]. Reproduced under the terms of the Creative Commons CC BY 4.0 license. Copyright 2021, Wiley-VCH. (**B**) Light actuation of a micro-structure with a photothermal effect generated by gold nanoparticles [60]. (**C**) LC microstructures actuated with light at different wavelengths depending on the dye used [61]. Reproduced under the terms of the Creative Commons CC BY 4.0 license. Copyright 2022, Wiley-VCH.

**Figure 8 micromachines-14-00124-f008:**
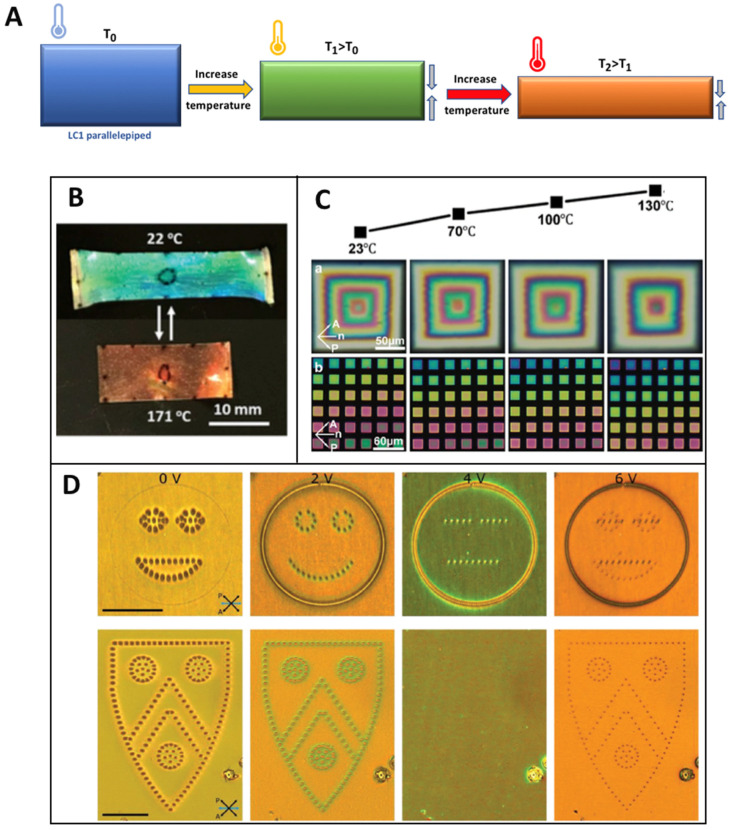
(**A**) Scheme of the change of color by the shrinking effect caused by thermal actuation. (**B**) LC color change principle with thermal actuation for a film example [65]. Reproduced under the terms of the Creative Commons CC BY-NC 4.0 license. Copyright 2020, Wiley-VCH. (**C**) Micro-squared structures that change color with the increase in temperature for the design of LOGOs [68]. Reproduced under the terms of the Creative Commons CC BY 4.0 license. Copyright 2020, Wiley-VCH. (**D**) Electric current for the reveal of LC micro-patterns [69]. Reproduced under the terms of the Creative Commons CC BY 4.0 license. Copyright 2018, Wiley-VCH.

## Data Availability

Not applicable.

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
