# Peer review of "Smart Nematic Liquid Crystal Polymers for Micromachining Advances"

_micromachines, 2023, doi:10.3390/mi14010124_

Round 1

Reviewer 1 Report

The authors reviewed recent developments related to micromachines made of liquid crystals and discussed further possible studies that can be useful to expand our studies in the field of micromachining. The concept of the review article is carefully thought out, however, after careful evaluation, I believe that additional discussion is necessary, including the following:

1.     In my opinion, abstract is too generic. Authors should include information what exactly the review is about. What issues are addressed in it, what form does it take, etc.

2.     I found typos in several places. I am asking the authors to thoroughly check and possibly correct the language of the article.

3.     What does the phrase ”We will focus on nematic liquid crystal only with an introduction on nematic LC behavior” mean?

4.     The authors discuss the nematic-to-isotropic phase transition under the influence of temperature. In this context, it is worth mentioning some recently published works: Materials 7.2 (2014): 1296-1317; Optics Express 26.3 (2018): 2443-2452; Advanced Optical Materials 7.6 (2019): 1801683, etc.

5.     Figure 6 is too large and contains too much information. I propose to simplify it so that its content becomes more readable and transparent. Alternatively, it can be divided into 2 smaller figures.

6.     The summary section is too sparse. It is extremely important that the authors try to outline both the problems that still need to be solved or optimized, and thoroughly discuss the future evolutions of this type of architectures.

Reviewer 2 Report

The review manuscript summarizes recent developments in micromachines made of liquid crystal and discuss further possible studies in the field of micromachining. Lots of previously reported works in this field were discussed in this manuscript. The authors introduced the molecular programming, activation methods, fabrication technics, and representative micromachines of liquid crystal micro-actuator. They particularly highlights two photon polymerization (TPP) fabrication technics. The review reported here is worth of publication in Micromachines. However, there are numerous concerns, which need to be carefully addressed prior to its acceptation for publication:

1. In most previously works, the substrate surfaces of LC cells are treated with PI or PAV polymer and then creating furrows by cloth. The reason for choosing these two polymers shall be added.

2. The configuration transition of azobenzene dopant in liquid crystal micro-actuator with different times (seconds, hours, days…), which depends on the chemicals used.  The authors should discuss the detailed dependency relationship between the configuration transition times and the chemicals.

3. Appearing for the first time, the full name of AM, FFF and DIW techniques should be shown.

4. The expression of Figure 2A, Figure 2C, Figure 6A may be better than  A of figure 2” ,  C of figure 2”, “A of figure 6” , and etc in the text. And the  corresponding description such as described in D offers, the last example E shows should be clearly indicated as described in Figure 2 D offers and the last example Figure 2E shows.

5. The resolution of Figure 2D and Figure 5 should be improved.

6.  “Different triggers allow to bend LC molecules as it is shown in figure” in the Line 139 on Page 4 should be “Different triggers allow to bend LC molecules as it is shown in Figure 3.

7. The relevant references with electric and magnetic actuation technics for LC molecular disorientation should be added.

8. The caption with “Figure 5. A) Different technics for the impression of object at the micro-scale.” should be  “Figure 5. A-C) Different technics for the impression of object at the micro-scale.”

9. Some repeated references, such as reference 2 and reference 3. And the format of references should be uniform.
